# Evaluating the Potential Safety Risk of Plant-Based Meat Analogues by Analyzing Microbial Community Composition

**DOI:** 10.3390/foods13010117

**Published:** 2023-12-29

**Authors:** Dan Hai, Baodang Guo, Mingwu Qiao, Haisheng Jiang, Lianjun Song, Ziheng Meng, Xianqing Huang

**Affiliations:** 1College of Food Science and Technology, Henan Agricultural University, Zhengzhou 450002, China; 15837184289@163.com (D.H.); guobaodang@126.com (B.G.); mingwu0309@163.com (M.Q.); slj69@126.com (L.S.); zihengm@163.com (Z.M.); 2Henan Engineering Technology Research Center of Food Processing and Circulation Safety Control, Zhengzhou 450002, China; m15890159032_1@163.com; 3Henan Shuanghui Investment & Development Co., Ltd., Luohe 462000, China; 4Henan Technology Innovation Center of Meat Processing and Research, Luohe 462000, China

**Keywords:** high-throughput sequencing, bacterial and fungal community, phenotype predicting, food safety

## Abstract

Plant-based meat analogues offer an environmentally and scientifically sustainable option as a substitute for animal-derived meat. They contribute to reducing greenhouse gas emissions, freshwater consumption, and the potential risks associated with zoonotic diseases linked to livestock production. However, specific processing methods such as extrusion or cooking, using various raw materials, can influence the survival and growth of spoilage and pathogenic microorganisms, resulting in differences between plant-based meat analogues and animal meat. In this study, the microbial communities in five different types of plant-based meat analogues were investigated using high-throughput sequencing. The findings revealed a diverse range of bacteria, including Cyanobacteria, Firmicutes, Proteobacteria, Bacteroidota, Actinobacteriota, and Chloroflexi, as well as fungi such as Ascomycota, Basidiomycota, Phragmoplastophyta, Vertebrata, and Mucoromycota. Additionally, this study analyzed microbial diversity at the genus level and employed phenotype prediction to evaluate the relative abundance of various bacterium types, including Gram-positive and Gram-negative bacteria, aerobic, anaerobic, and facultative anaerobic bacteria, as well as potential pathogenic bacteria. The insights gained from this study provide valuable information regarding the microbial communities and phenotypes of different plant-based meat analogues, which could help identify effective storage strategies to extend the shelf-life of these products.

## 1. Introduction

Meat of animal origin has traditionally been a significant source of protein in human diets [1]. However, animal-origin meat production is accompanied by greenhouse gas emissions and requires significant water consumption. For instance, the production of 1 kg of beef is responsible for releasing 26.5 kg of CO_2_ and consuming 15,000 L of water [2,3]. As society develops and the population grows, the explosive increase in meat consumption could lead to a surge in the demand for meat production, exacerbating greenhouse gas emissions and freshwater consumption. This raises concerns regarding the sustainability of meat production. Meanwhile, in terms of human health, meat consumption increases the risk of heart disease [4,5], and livestock farming leads to the propagation of zoonotic diseases, such as swine flu and brucellosis [6,7]. Consequently, there is a need to explore sustainable alternatives to animal meat. Substituting animal meat with meat analogues presents a scientifically supported and sustainable approach [8]. Besides sustaining enough protein, meat analogues reduce greenhouse gas emissions, lower freshwater consumption, and eliminate the associated risk of chronic and zoonotic diseases [9]. Nowadays, meat analogues mainly comprise plant-based meat and cultured meat [10]. Compared to cultured meat, plant-based meat has a longer history and mature craftsmanship. As a result, plant-based meat is more readily accepted by consumers and tends to be more affordable [11,12].

Usually, in the production of plant-based meat analogues, plant-derived raw materials such as proteins, lipids, and carbohydrates are mixed with added vitamins, minerals, additives, and other ingredients [13,14,15,16]. And subsequent processing with cooking or extrusion at high temperatures endows these meat analogues with the mimicked taste, texture, and nutritional profile of animal-origin meat [13,17]. Therefore, there are significant differences in microbial composition between plant-based meat analogues and animal meat. Furthermore, the differences in nutritional profile, pH, and texture may influence adhesion, invasion, and proliferation, ultimately impacting the survival and growth of both spoilage and pathogenic microorganisms [18,19,20]. While significant efforts have been made to ensure the safety of microbial communities in animal-origin meats [21], greater attention should also be given to plant-based meat analogues due to their distinct nutritional profiles and processing steps. According to the reports, microorganisms in samples containing meat analogues seem to proliferate faster [22], and meat spoilage and pathogenic microorganisms grow at abused storage temperatures (22 °C and 32 °C) on ground beef and plant-based meat analogues, indicating the potential microbial safety and quality challenges associated with plant-based meat analogues [23].

Thus, the objective of this study is to evaluate the food safety of plant-based meat analogues in the aspect of microbial quality via high-throughput sequencing analysis in five different types of plant-based meat analogues. By detecting spoilage and pathogenic bacteria, this research aims to provide insights into controlling the spoilage of plant-based meat analogues and ensuring their safety.

## 2. Materials and Methods

### 2.1. Materials and Reagents

The One-Step DNA Extraction Kit was obtained from Tianenze Gene Technology Co., Ltd., Beijing, China. The E.Z.N.A.^®^ Bacterial Genomic DNA Extraction Kit was purchased from Omega Bio-Tek, Norcross, GA, USA. The TransStart Fastpfu DNA Polymerase was obtained from Transgene Biotechnology Co., Ltd., Beijing, China.

### 2.2. Sample Preparation

The planted-based meat samples, including beef jerky, vegetable sausage, beef grains, crab mayonnaise, and beef grain billet, were collected from the plant-based meat analogues company and labeled as BJ, VS, BG, CM, and BGB, respectively. The samples were directly obtained from the company, chilled at −20 °C, and cut into uniform shape. For each kind of plant-based meat analogue, three replicate samples from different plant-based meat samples were prepared. The main content of these plant-based meat analogues is listed in Table 1.

### 2.3. Bacterial and Fungal Genomic DNA Extraction

The planted-based meat samples were placed in sterile homogenization bags and homogenized for 120 s in a stomacher after adding 225 mL of the sterilized saline solution. A 20 mL aliquot of the homogenized mixture was collected via centrifugation at 17,226× *g* for 10 min, and the bacteria and fungi were separated, respectively. Then, bacterial and fungal genomic DNA was extracted using a bacterial genomic DNA extraction kit and fungal genomic DNA extraction, respectively.

### 2.4. High-Throughput Sequencing

The total genomic DNA used to analyze the difference in microbial quality in the five kinds of plant-based meat analogues was extracted in accordance with the manufacturer’s protocol, and the DNA quality was determined using 1% agarose gel electrophoresis. The DNA concentration and purity were measured using a UV-Vis Spectrophotometer (Nanodrop 2000C, Thermo, Waltham, MA, USA). Three repeats were conducted for each sample.

The microbial communities, including bacterial and fungal composition, were analyzed via high-throughput sequencing, respectively. For the bacterial composition analysis, the V3–V4 regions of the 16S rRNA gene were amplified with the primers 338F (ACTCCTACGGGAGGCAGCAG) and 806R (GGACTACHVGGGTATCTAAT) to determine the bacterial species. To analyze the fungal composition, the V5–V7 regions of ITS were amplified with 817F (TTAGCATGGAATAATRRAATAGGA) and 1196R (TCTGGACCTGGTGAGTTTCC). The targeted DNA was amplified using the TransStart Fastpfu DNA Polymerase and purified via the AxyPrepDNA extraction kit. A system including 0.2 μM of forward and reverse primers and approximately 10 ng of templated DNA was conducted. The instrument used for amplifying the targeted gene was the ABI GeneAmp&reg 9700 series. The thermal cycling procedure consisted of initial denaturation at 98 °C for 60 s, followed by 30 cycles of denaturation at 98 °C for 10 s, annealing at 50 °C for 30 s, and elongation at 72 °C for 60 s, and a final extension at 72 °C for 300 s. The PCR product quality was determined using 1% agarose gel electrophoresis and quantified by the QuantiFluor™ -ST system (Promega, Madison, WI, USA). The DNA libraries were prepared with a TruSeq^TM^ DNA Sample Prep Kit (Illumina Inc., San Diego, CA, USA) in accordance with its manufacturer’s protocol and sequenced using Illumina Hiseq 2500 system, generating 250 bp paired-end reads. The process mainly contained the following: one end of the DNA fragment was complementary to the primer bases and immobilized on the chip. Using the DNA fragment as a template and the primer sequences immobilized on the chip, PCR synthesis was performed to generate the target DNA fragment to be tested on the chip. After denaturation and annealing, the other end of the DNA fragment on the chip was randomly bound to another nearby complementary primer, forming a “bridge” that was also immobilized. PCR amplification was carried out to generate DNA clusters. The amplified DNA fragments were linearized into single strands. Modified DNA Polymerase and fluorescently labeled dNTPs were added. Each base was incorporated during each cycle of synthesis. The reaction plate surface was scanned with lasers to read the types of nucleotides incorporated during the first cycle of each template sequence. The “fluorescent group” and “terminator group” were chemically cleaved, allowing the restoration of the sticky 3′ ends for the incorporation of the next nucleotide. The fluorescent signals collected in each round were analyzed to determine the sequence of the template DNA fragment.

### 2.5. Date Processing and Bioinformatics Analysis

For sequenced data, the paired-end sequences were merged according to the overlap sequence using FLASH (Version 1.2.11) and were filtered via QIIME (Version 1.9.1). High-quality sequences were aligned in accordance with the SILVA (Version 138) alignment and UNITE alignment via an RDP classifier (Version 2.13) and USEARCH (Version 11), respectively. And the sequences with 97% similarity or higher were clustered into operational taxonomic units (OTUs) using UPARSE (Version 11). The taxonomy analysis of the 16S rRNA sequence and ITS sequence was conducted against the SILVA (Version 138) 16S rRNA database and the UNITE (Version 8.0) fungal ITS database with a confidence threshold of 0.7, respectively. Furthermore, the richness and diversity of the microbial community were evaluated by analyzing the alpha-diversity indices using MOTHUR (Version 1.30.2), including Good’s coverage, the Chao 1 estimator, Simpson, Shannon, and Sobs indexes. The beta diversity analysis, including principal co-ordinates analysis (PCoA) and a hierarchical heatmap, was conducted to analyze the difference in microbial communities between the fifteen samples. 

## 3. Results

### 3.1. High-Throughput Sequencing Results and Diversity Analysis

In this study, 1,397,628 valid reads were obtained for bacteria and classified into 2395 operational taxonomic units (OTUs) with a 97% similarity threshold or higher after merging and filtering the fifteen samples. The number of reads per sample ranged from 75,721 to 102,304, with an average length of 416 base pairs (bp). For fungi, a total of 1,443,457 valid reads with an average length of 382 bp and sequence numbers ranging from 79,115 to 105,603 per sample were obtained through the quality control from 1,551,511 raw reads. The fungal reads were classified into 403 OTUs using the same criteria. The alpha diversity indexes of Ace, Chao, Good’s coverage, Shannon, Simpson, and Sobs were employed to evaluate the richness and diversity of the bacterial and fungal communities and their values, along with the sequence number, average length, and OTU number, are listed in Appendix A, respectively. Rarefaction curves were used to represent diversity as a function of the library size [24], and all the rarefaction curves of the samples showed a clear trend toward the saturation plateau, indicating that the sequencing covered the majority of microbial members. Consequently, the reliable and representative microbial diversity in this study is supported by the obtained data (Appendix A).

For bacteria, Good’s coverage of each sample ranged from 99.69% to 99.90%, indicating that almost all bacteria in the plant-based meat analogues could be detected. The Ace and Chao indexes reflected the species richness of the bacterial community, and the Shannon index represented the species abundance as distributed among all species in the community. These three indexes, Ace, Chao, and Shannon in CM samples, were the highest compared to other samples, representing the highest richness and diversity.

Good’s coverage for fungi from 99.94% to 99.98% indicates that all fungi present in the samples could be detected. The Ace and Chao indexes were highest in the BG samples, indicating the greater richness of fungal species in those samples compared to the BJ and BGB samples. The Shannon index was also highest in the BG samples, indicating a greater species abundance in the fungal community of those samples, while the CM samples showed the lowest value for this index.

### 3.2. Taxonomic Composition of Microbial Community

For the bacterial composition in the fifteen samples, a total of 2395 OTUs were classified into 40 phyla, 117 classes, 248 orders, 420 families, 872 genera and 1412 species. At the phylum level, there were three dominant phyla: Cyanobacteria, Firmicutes, and Proteobacteria. These three phyla accounted for 95% of the total sequences in all fifteen samples (Figure 1A).

In the BJ group, Cyanobacteria was the most dominant phylum, comprising approximately 94% of the sequences. Firmicutes accounted for 1.93%, and Proteobacteria made up 2.96% of the total sequences. In the CM group, Firmicutes was the most predominant phylum, representing 83.3% of the sequences. Cyanobacteria, and Proteobacteria contributed 4.39% and 3.34% of the total sequences, respectively. The percentages of Cyanobacteria as the dominant phylum in the other groups ranged from 32.49% to 76.61%, while the occupancy of Firmicutes varied from 9.23% to 54.46%.

At the genus level, the dominant genera in all fifteen samples were *norank_f_norank_o_Chloroplast*, Klebsiella, Acinetobacter, and *norank_f_Mitochondria*, which accounted for over 90% of the sequences (Figure 1B). However, there were noticeable differences in their proportions among the five groups. In the CM group, the genus *norank_f_norank_o_Chloroplast* had a low proportion of 3.94%, while it ranged from 32.45% to 94.60% in the VS, BJ, BG, and BGB groups. The four most dominant genera in the CM group were *Weissella*, *Lactobacillus*, *Leuconostoc*, and *Acinetobacter*, with contributions of 36.57%, 23.75%, 11.33%, and 1.23%, respectively. In the BJ, CM, and BG groups, the genus Thermoactinomyces accounted for less than 1% of the total reads, with proportions of 0.01%, 0.2%, and 0.039%, respectively. In contrast, the VS and BGB groups had a much higher percentage of *Thermoactinomyces*, with occupancies of 53.15% and 31.99%, respectively.

For the fungal community, the 403 OTUs in a total of fifteen samples were classified into 22 phyla, 56 classes, 84 orders, 124 families, 145 genera, and 170 species. At the phylum level, the dominant phyla were Ascomycota, Basidiomycota, and Phragmoplastophyta, which, together, accounted for 98.63% of the total reads in the fifteen samples (Figure 2A). In groups BJ, BG, and BGB, Ascomycota showed the highest proportion, accounting for 78.16%, 79.09%, and 85.53%, respectively. In contrast, the dominant phylum in groups VS and CM was Basidiomycota, contributing 76.42% and 89.8%, respectively. Phylum Phragmoplastophyta, with a lower abundance, was found in groups BJ, BG, and BGB, with proportions of 10.63%, 0.71%, and 1.62%, respectively. Additionally, another phylum, Vertebrata, was found in group BG with a proportion of 5.02%, which was lower than 0.01% in the other four groups.

At the genus level, the most dominant genera were Pleurotus, Aspergillus, *Cladosporium*, and *Boeremia*, with proportions of 2.87%, 26.64%, 16.98%, and 6.14% in the fifteen samples, respectively (Figure 2B). In groups VS and CM, Pleurotus was the dominant phylum with significantly higher abundance, contributing 76.21% and 59.93%, which may be ascribed to the raw materials in the VS and CM groups. The second most dominant genus in the VS group was *unclassified_f_Saccharomycetaceae*, with a proportion of 22.06%. The second most dominant genus in the CM group was *Apiotrichum* (29.34%), which was lower than 2.43% in the other four groups. In contrast, the fungal communities in groups BJ, BG, and BGB were more diverse. The proportion of *Aspergillus* varied from 28.66% to 1.79%, with the highest proportion observed in group BG, followed by group BJ. Genus *Boeremia*, which was a pathogenic bacterium causing rotten leaves disease, mainly distributed in groups BJ, BG, and BGB, accounted for 0.59%, 5.99%, and 1.10%, respectively.

### 3.3. The Similarity of Microbial Community

Hierarchical cluster analysis was employed to analyze the similarity of microbial communities in the fifteen samples. The bacterial heatmap analysis, based on the top 50 abundant bacterial genera (Figure 3A), revealed that groups BJ and BG formed one cluster and clustered with group BGB, while group CM showed a clear difference from the other groups. This was further confirmed by the PCoA results in the beta diversity analysis (Figure 4A). The PCoA results indicated that the first principal component accounted for 50.77% of the total variation, while the second principal component accounted for 31.17% of the variation. Group CM samples were separated from the others along the PC1 axis, and group VS samples were separated along the PC2 axis. Additionally, group BGB samples exhibited lower repeatability. Furthermore, the Venn diagram (Figure 5A) was used to illustrate the differences in bacterial communities among the samples. It showed that 180 OTUs were shared among all the samples, representing 19.56%, 17.09%, 7.33%, 12.41%, and 8.47% of the total OTUs in VS, BJ, CM, BG, and BGB group samples, respectively. The proportion of unique OTUs in each group was 5.43%, 6.07%, 21.34%, 7.24%, and 15.07% for the VS, BJ, CM, BG, and BGB groups, respectively. 

For fungi, the heatmap analysis was performed with the top 50 abundant fungal genera (Figure 3B). Groups CM and VS were clustered together, while groups BGB, BJ, and BGB formed another cluster. The results were confirmed using the PCoA analysis, indicating that groups CM and VS had similar fungal communities. The PCoA results showed that 57.76% of the total variations were attributed to the first principal component, and 17.26% of the total variations belonged to the second principal component (Figure 4B). Groups VS and CM were grouped to the left of the graph along the PC1 axis, while group BJ and group BG were grouped along the PC2 axis. The Venn diagram indicates that the five groups shared 63 OTUs in common, which contributed to 29.3%, 10.16%, 35.39%, 8.77%, and 11.07%, respectively (Figure 5B). Moreover, the VS group shared no unique OTU with other groups, while the unique OTU in the BJ, CM, BG, and BGB group accounted for 5.32%, 5.05%, 11.28%, and 4.56%.

### 3.4. The Relative Abundance of Gram-Positive and Gram-Negative Bacteria

To further analyze the community in the fifteen samples, phenotype prediction at the genus level was employed to analyze the Gram-positive and Gram-negative bacteria composition. As shown in Figure 6, groups VS, CM, and BGB were predominantly composed of Gram-positive bacteria, while Gram-negative bacteria were dominant in groups BJ and BG. Specifically, group CM had *Weissella*, *Lactobacillus*, and *Leuconostoc* as the dominant Gram-positive bacteria. In groups VS and BGB, Thermoactinomyces was the dominant Gram-positive bacterium, while Weissella, *Lactobacillus*, and *Staphylococcus* were present in smaller proportions in group BGB. The Gram-negative bacteria *norank_f_norank_o_Chloroplast* and *norank_f_Mitochondria* were detected and dominant in groups BJ and BG. Additionally, the Gram-negative bacteria Klebsiella showed a proportion of 19.32% in group BG.

### 3.5. The Relative Abundance of the Aerobic, Anaerobic, and Facultative Anaerobic Bacteria

Storage condition packaging atmospheres have a significant impact on microbial growth, and many modified atmosphere packaging methods have been developed to extend the shelf-life of food [25,26,27]. Thus, the compositions of the aerobic, anaerobic, and facultative anaerobic bacteria at the genus level in the fifteen samples were analyzed. The aerobic bacteria in the samples mainly contained *Lactobacillus*, *Acinetobacter*, *Brevundimona*, *Pseudomonas,* and *Comamonas* (Figure 7A). Among them, *Lactobacillus*, *Acinetobacter,* and *Pseudomonas* were the spoilage bacteria found in animal meat products [28]. However, the aerobic bacteria in the VS group accounted for a lower proportion. As shown in Figure 7B, the results of the anaerobic bacteria composition in the samples show that *Thermoactinomyces* was dominant in group VS and BGB, accounting for 75.69% and 47.56%, respectively. However, the proportions of anaerobic bacteria in groups BJ, CM, and BG were minor. Finally, the relative abundance of facultative anaerobic bacteria in the samples was analyzed (Figure 7C). The results show that the proportions of facultative anaerobic bacteria in each group were quite different. The abundance of facultative anaerobic bacteria in groups CM, BG, and BGB was higher than the other two groups. *Leuconostoc*, *Klebsiella*, and *Staphylococcus* were the major facultative anaerobic bacteria in groups CM, BG, and BGB. In conclusion, the analysis of the proportions of the aerobic, anaerobic, and facultative anaerobic bacteria in each sample can provide guidance for the packaging atmospheres of plant-based meat analogues and prolong the shelf-life.

### 3.6. The Relative Abundance of the Potential Pathogenic Bacteria

Nowadays, food safety caused by pathogenic bacteria is a public concern. Therefore, the relative abundance of potentially pathogenic bacteria in these five plant-based meat analogues has been studied (Figure 7D). The potential pathogens in the samples were mainly *Thermoactinomyces*, *Klebsiella*, *Staphylococcus*, *Acinetobacter*, and *Pseudomonas*. The pathogenic bacteria *Thermoactinomyces* were dominant in groups VS and BGB, accounting for 75.69% and 47.56%, which was lower in other groups. The pathogenic bacteria *Klebsiella*, mainly causing Klebsiella pneumoniae infection [29], was detected in all samples at different proportions. In groups BG and BGB, *Klebsiella* bacteria were identified at 3.83% and 1.15%, while *Klebsiella* bacteria in other groups were much lower than 0.25%. Additionally, another pathogenic bacteria, Acinetobacter, which is a type of bacteria that is resistant to many types of antibiotics [30], was detected in all samples, accounting for 0.21% (VS group), 0.59% (BJ group), 1.23% (CM group), 2.57% (BG group), and 2.55% (BGB group). A potential pathogenic bacterium, *Staphylococcus*, which may cause suppurative disease processes in humans and animals [31], was detected in group BGB (5.92%), and the proportions in other groups were lower than 0.50%. The detection of pathogenic bacteria indicated that more attention should be paid to the safety of plant-based meat analogues.

## 4. Discussion

Plant-based meat analogues offer an environmentally and scientifically sustainable option as a substitute for animal-derived meat. They contribute to reducing greenhouse gas emissions, freshwater consumption, and the potential risks associated with zoonotic diseases linked to livestock production. However, specific processing methods, such as extrusion or cooking, using various raw materials can influence the survival and growth of spoilage and pathogenic microorganisms, resulting in differences between plant-based meat analogues and animal meat. Hence, it is necessary to explore the microbial communities in different types of plant-based meat analogues.

The microbial communities, including bacteria and fungi, and phenotype prediction in the five kinds of plant-based meat analogues were analyzed using high-throughput sequencing. For bacteria at the genus level, *norank_f_norank_o_Chloroplast, Klebsiella*, *Acinetobacter*, and *norank_f_Mitochondria* were the dominant bacteria, and the proportions of these bacteria in each group were different. However, *norank_f_norank_o_Chloroplast* and *norank_f_Mitochondria* in the bacterial community originate from the raw materials shown in Table 1. For fungi at the genus level, *Pleurotus*, *Aspergillus*, *Cladosporium,* and *Boeremia* were the most dominant genera. The reason for the diversity of the microbial composition may be attributed to the various raw materials and complicated cultural processes used. For instance, oyster mushrooms in the CM and VS group resulted in a high proportion of *Pleurotus* genera. Subsequent processing with cooking or extrusion results in the recontamination of this plant-based meat analogue [18]. The differences in texture of these plant-based meat analogues, such as CM, which was a liquid with a higher water content than other samples, may be another reason for the clear difference in the microbial composition.

Food safety is an important aspect of consumer concerns. However, only a few data are available about plant-based meat analogues. The food safety risk of meals prepared with plant-based meat analogues is slightly higher than their meat-containing counterparts [22]. Although most of these bacteria can be killed by the heat generated during the extrusion process, some spore-forming bacteria, such as *Clostridium* spp. or *Bacillus* spp., may survive the heating process [32]. In a study, spores of Clostridium sporogenes (ATCC 19404) in protein ingredients were inactivated to undetectable levels by the extrusion process, while *Bacillus amyloliquefaciens* (AB255669 and LMA008A; <1000 CFU/g) was detected in the final protein extrudate, possibly due to post-extrusion recontamination [18,32]. Hence, phenotype prediction was conducted to reveal the relative abundance of spoilage and pathogenic bacteria in five types of plant-based meat analogues. Gram-positive and Gram-negative bacteria, aerobic, anaerobic, facultative anaerobic bacteria, as well as potential pathogenic bacteria in the five types of plant-based meat analogues, were evaluated.

## 5. Conclusions

An in-depth analysis of the microbial composition of five types of plant-based meat analogues samples was performed using high-throughput sequencing technology. This analysis provides valuable information on the diversity and compositional changes in plant-based meat analogues. However, there are some areas for improvement in a later study, such as more samples to ensure the accuracy of data, more kinds of plant-based meat analogues, the relationship between microbial composition and texture, and the pH value. Hence, further research directions include the functional analysis of microbial communities associated with raw materials and specific processing methods to prolong the product’s shelf-life and reduce the proliferation of pathogenic bacteria to ensure food safety.

## Figures and Tables

**Figure 1 foods-13-00117-f001:**
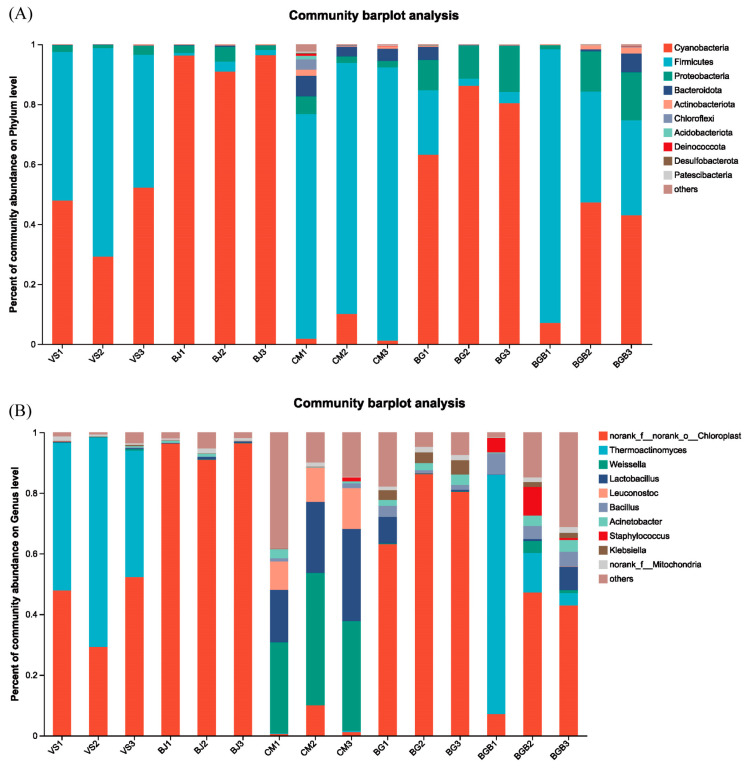
Relative abundance of the bacterial community in the fifteen samples. (**A**) At the phylum level; (**B**) At the genus level.

**Figure 2 foods-13-00117-f002:**
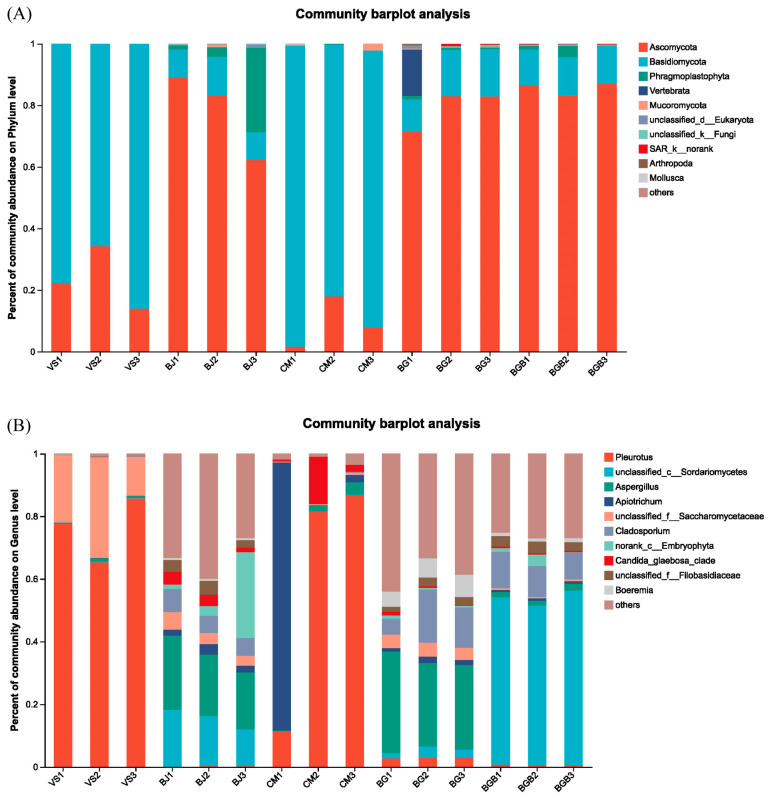
Relative abundance of the fungal community in the fifteen samples. (**A**) At the phylum level; (**B**) At the genus level.

**Figure 3 foods-13-00117-f003:**
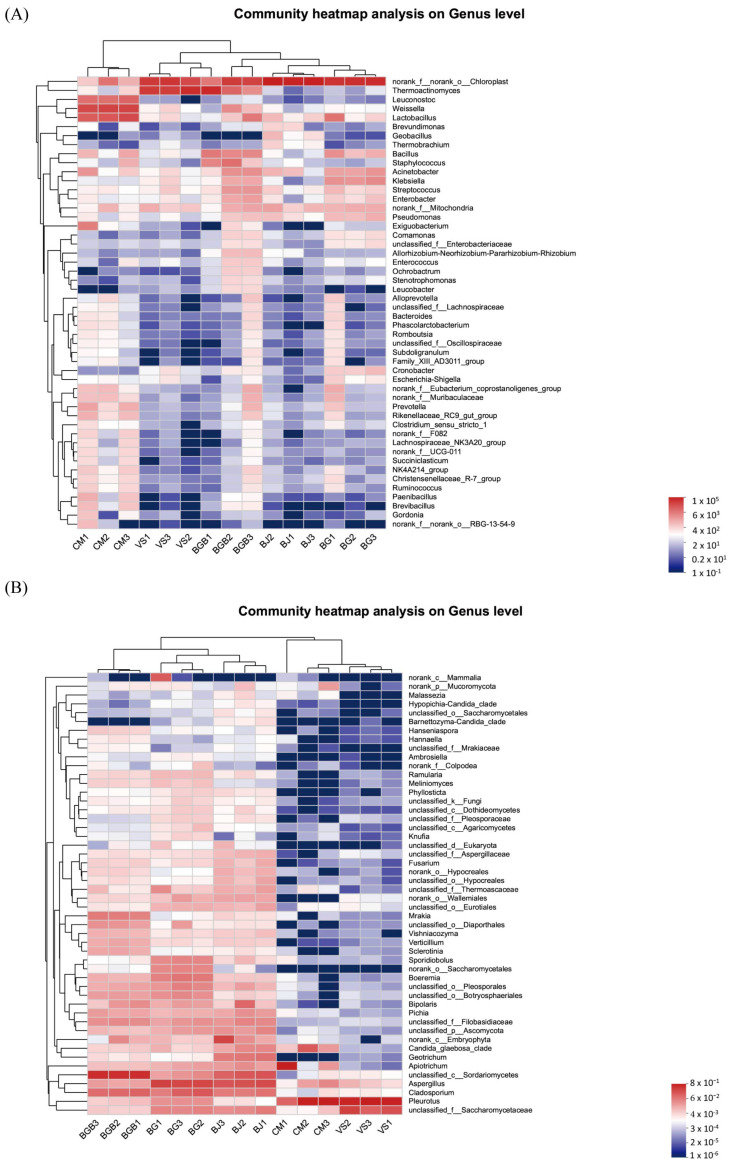
Heatmap analysis of the microbial community of the fifteen plant-based meat analogues. (**A**) Heatmap analysis of the bacterial community; (**B**) Heatmap analysis of the fungal community. The heatmap only showed the relative abundance of the top 50 genera of the bacterial and fungal communities, respectively.

**Figure 4 foods-13-00117-f004:**
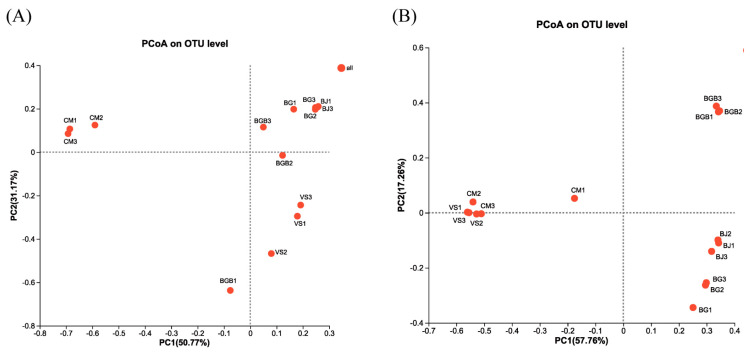
Principal co-ordinates analysis of the microbial community of the fifteen plant-based meat analogues. (**A**) Principal co-ordinates analysis of the bacterial community; (**B**) Principal co-ordinates analysis of the fungal community.

**Figure 5 foods-13-00117-f005:**
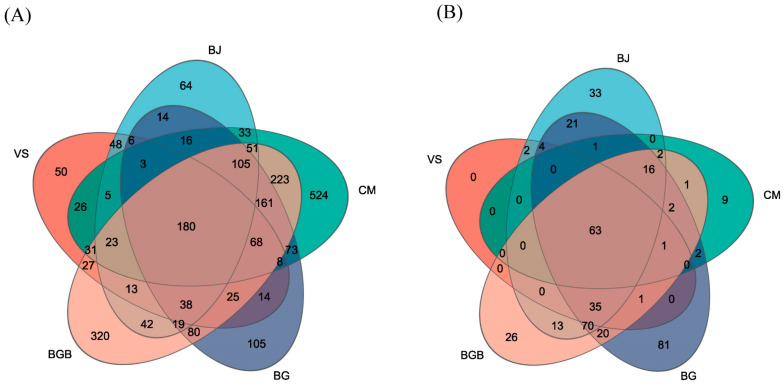
Venn diagram analysis of the number of shared and unique OTUs in the samples. (**A**) The shared and unique OTUs for bacterial communities in the samples; (**B**) The shared and unique OTUs for fungal communities in the samples.

**Figure 6 foods-13-00117-f006:**
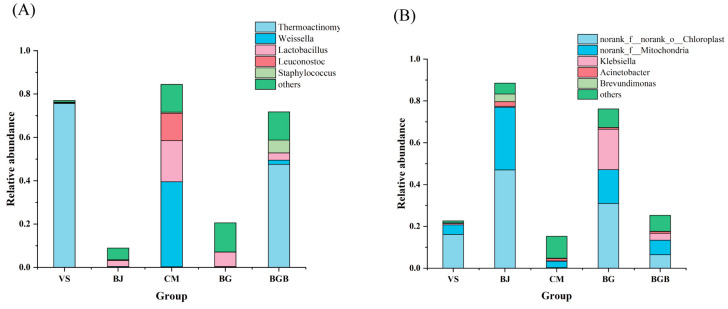
The relative abundance of the Gram-positive and Gram-negative bacteria. (**A**) The Gram-positive bacteria; (**B**) The Gram-negative bacteria.

**Figure 7 foods-13-00117-f007:**
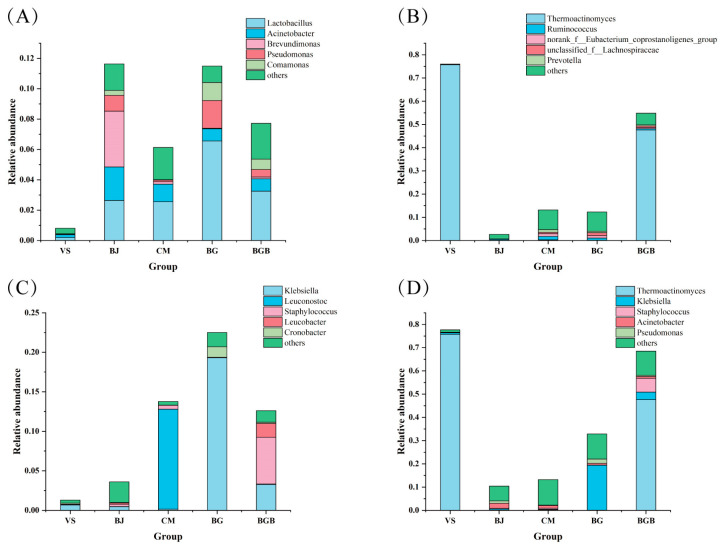
The relative abundance of aerobic, anaerobic, and facultative anaerobic bacteria. (**A**) The aerobic bacteria; (**B**) The anaerobic bacteria; (**C**) The facultative anaerobic bacteria; (**D**) Potential pathogens.

**Table 1 foods-13-00117-t001:** Main content of the five kinds of plant-based meat analogues.

Plant-Based Meat Analogues	Main Content
Beef jerky	soy brushed protein, water, golden sugar, soy protein isolate, soybean oil, brewed soy sauce, flavor, spices, lactic acid, salt, malt extract, round psyllium husk, yeast extract, methylcellulose, gum arabic, citrus fiber, soy protein powder
Vegetable sausage	vegetable oil, water, oyster mushroom, soy tissue protein, edible salt, monosodium glutamate, food flavor, white vinegar, carotene, gum arabic, ginger powder, methylcellulose, soy protein powder, guar gum, locust bean gum.
Beef grains	soy brushed protein (low-temperature desolventizing edible soybean meal, gluten, soy protein isolate, cornstarch), water, trehalose, spicy beef flavored compound sauce, soy protein isolate, soybean oil, round bud psyllium husk, malt extract, lactic acid
Crab mayonnaise	vegetable oil, water, oyster mushroom, soy tissue protein, edible salt, monosodium glutamate, food flavor, white vinegar, carotene, gum arabic, ginger powder, methylcellulose, soy protein powder, guar gum, locust bean gum.
Beef grains billet	soy brushed protein (low-temperature edible soybean meal, gluten powder, soy protein isolate, cornstarch), water, trehalose, spicy beef flavor compound sauce, soy protein isolate, soybean oil, round bud psyllium husk, malt extract, lactic acid

## Data Availability

The raw data are available in the public, community-supported repository with an accession number for these SRA data: PRJNA990984, Temporary Submission ID: SUB13604823.

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
