# Peer review of "Evaluating the Potential Safety Risk of Plant-Based Meat Analogues by Analyzing Microbial Community Composition"

_foods, 2023, doi:10.3390/foods13010117_

Round 1

Reviewer 1 Report

Comments and Suggestions for Authors

The manuscript entitled "Evaluating potential safety risk of plant-based meat analogues by analyzing microbial community composition", submitted by Hai et al., evaluated the microbial quality of five different types of plant-based meat analogues by high-throughput sequencing analysis. The paper is well-written and logistically organized, but the authors must address several areas for improvement in the manuscript.

Lines 41-42. This reviewer disagrees with the authors. There is no clear association between meat consumption and increased risk of heart disease, diabetes, and hypertension. The evidence only suggests an association between MEAT PRODUCTS and an increased risk of heart disease (as mentioned in reference 5 given by the authors).

Lines 81-86. a) It is unclear if the authors used only one sample and subsequently divided it into three parts or used three different Plant-based meat samples. b) were the samples fresh? Or were the samples taken close to the best-before date? Please include this information.

Lines 169-175. Please specify which means J, L, G, and P samples.

Lines 351-352. Please extend the discussion about the reasons for the diversity of microbial composition.

Lines 357-359. Please describe some published studies about microbial composition (including pathogen growth risk) in Plant-based meat products

Lines 360-366. This reviewer suggests a conclusion section. Please include limitations of the study, for example, the small number of samples and others that the authors consider appropriate.

Line 384. Please update some of the references used in all the manuscript.

I hope my comments help to improve the manuscript.

Author Response

Q1. Lines 41-42. This reviewer disagrees with the authors. There is no clear association between meat consumption and increased risk of heart disease, diabetes, and hypertension. The evidence only suggests an association between MEAT PRODUCTS and an increased risk of heart disease (as mentioned in reference 5 given by the authors).

A1: Thanks for your advice. we have modified the inappropriate description in lines 41-42 as follow: Meanwhile, in terms of human health, meat consumption increases the risk of heart disease.

Q2. Lines 81-86. a) It is unclear if the authors used only one sample and subsequently divided it into three parts or used three different Plant-based meat samples. b) were the samples fresh? Or were the samples taken close to the best-before date? Please include this information.

A2: Thanks for your advice. We have modified the description of sampling method to be more accurate in the manuscript. a) the samples were prepared from three different plant-based meat samples. b) the samples were directly obtained from the company and stored at -20℃。

Q3. Lines 169-175. Please specify which means J, L, G, and P samples.

A3: Thanks for your correction. We have modified the errors in Lines 169-175.

Q4: Lines 351-352. Please extend the discussion about the reasons for the diversity of microbial composition.

A4: Thanks for your suggestion. We have extended the discussion about the reasons for the diversity of microbial composition in the manuscript as follow: The reason for the diversity of the microbial composition may be attributed to the various raw materials and the complicated cultural processes. For instance, oyster mushroom in CM and VS group resulted in the high proportion of Pleurotus genera. The subsequent processing with cooking or extrusion will result in the recontamination of the plant-based meat analogues [18]. The differences in texture of these plant-based meat analogues, such as CM, which was liquid with higher water content than other samples, may be another reason for the obvious difference in the microbial composition.

Q5: Lines 357-359. Please describe some published studies about microbial composition (including pathogen growth risk) in Plant-based meat products.

A5: Thanks for your advices. We have described some published studies about microbial composition in plant-based meat products as follow: The food safety risk of meals prepared with plant-based meat analogues is slightly higher than their meat-containing counterparts [22]. Although most of these bacteria can be killed by the heat generated during the extrusion process, some spore-forming bacteria, such as Clostridium spp. or Bacillus spp., may survive the heating process [32]. In a study, spores of Clostridium sporogenes (ATCC 19404) in protein ingredients were inactivated to undetectable levels by the extrusion process, while Bacillus amyloliquefaciens (AB255669 and LMA008A; <1000 CFU/g) was detected in the final protein extrudate, possibly due to post-extrusion recontamination [18,32].

Q6: Lines 360-366. This reviewer suggests a conclusion section. Please include limitations of the study, for example, the small number of samples and others that the authors consider appropriate.

A6: Thanks for your suggestions. We have added the conclusion section as follow: In-depth analysis of the microbial composition of five types of plant-based meat analogues samples was performed using high-throughput sequencing technology. This analysis provides valuable information on the diversity and compositional changes of plant-based meat analogues. However, there are some areas for improvement in later study, such as more samples to ensure the accuracy of data, more kinds of plant-based meat analogues, the relationship between microbial composition and texture and pH value. Hence, further research directions include functional analysis of microbial communities associated with the raw materials and specific processing methods to prolong the product shelf-life, and reduced the proliferation of the pathogenic bacteria to ensure the food safety.

Q7: Line 384. Please update some of the references used in all the manuscript.

A7: Thanks for your suggestions. We have updated the references in the manuscript.

Reviewer 2 Report

Comments and Suggestions for Authors

This is an interesting study and the authors have collected some dataset using cutting edge methodology. Below I have provided some remarks on the manuscript.

1. Line 35. the production animal origin meat production. Check this sentence. It doesn’t make sense. Please revise.

2. Line 43…… “Livestock farming leads to the propagation of zoonotic diseases, such as avian influenza, swine flu and brucellosis”. Basically, the term livestock refers to farm animals with the exception of poultry. Avian influenza is mainly for poultry. Please revise this statement.

3. There are some typos encountered throughout the manuscript. Some of these errors are listed below. 

-Line 46. Besides of enough protein

-Line 55, 138. And subsequent processing treatment

- Line 338… live-stock production. Livestock one word.

-Line 342………. Plant-based ………. “P” ….…should be small.

- Line 349……. For fungi, At the genus level………………. ‘’At’’ …. A should be small.

4. Line 66…... at abused storage temperatures. For the sake of clarity, could you please specify this temperature using parentheses or round brackets?

5. Line 69…... to evaluate the food safety of………….the term "food" is not  necessary as you have mentioned plant-based meat as a food type.

6. Line 81. Please revise the sentence “The planted-based meat samples such as beef jerky, vegetable sausage, beef grains, crab mayonnaise, and beef grains billet were collected from”.

7. Line 92…. to collect the bacteria and fungi, respectively. In this sentence the term respectively indicates there is something missing. Please revise this sentence.

8. Line 92-93. bacterial genomic DNA was extracted using a bacterial genomic DNA extraction kit and fungal genomic DNA extraction, respectively. This statement is confusing, please revise.

9. Line 312 - 315. Redundancy of “The relative abundance of”. Please revise.

10. Please separate the conclusion part from the discussion, if possible, provide future directions along with it as well.

Thank you,

Best wishes

Comments on the Quality of English Language

I have encountered some grammatic and typographic errors throughout the manuscript. Hence, the paper should undergo professional language editing before it can be published.

Thank you,

Best wishes

Author Response

Q1. Line 35. the production animal origin meat production. Check this sentence. It doesn’t make sense. Please revise.

A1: Thanks for your suggestion. We have modified the sentence as follow: the animal origin meat production is accompanied by greenhouse gas emissions and requires significant water consumption.

Q2. Line 43…… “Livestock farming leads to the propagation of zoonotic diseases, such as avian influenza, swine flu and brucellosis”. Basically, the term livestock refers to farm animals with the exception of poultry. Avian influenza is mainly for poultry. Please revise this statement.

A2: Thanks for the reviewer’s suggestions. We have modified the description as follow: and the livestock farming leads the propagation of zoonotic diseases, such as swine flu and brucellosis.

Q3. There are some typos encountered throughout the manuscript. Some of these errors are listed below.

-Line 46. Besides of enough protein

-Line 55, 138. And subsequent processing treatment

- Line 338… live-stock production. Livestock one word.

-Line 342………. Plant-based ………. “P” ….…should be small.

- Line 349……. For fungi, At the genus level………………. ‘’At’’ …. A should be small.

A3: Thank you very much for the correction. We have corrected the errors and  revised the manuscript.

Q4. Line 66…... at abused storage temperatures. For the sake of clarity, could you please specify this temperature using parentheses or round brackets?

A4: Thanks for your suggestion. We have listed the temperature in the parentheses in Line 66.

Q5. Line 69…... to evaluate the food safety of………….the term "food" is not  necessary as you have mentioned plant-based meat as a food type.

A5: Thanks for your suggestion. We have deleted the term “ food“ in Line 69.

Q6. Line 81. Please revise the sentence “The planted-based meat samples such as beef jerky, vegetable sausage, beef grains, crab mayonnaise, and beef grains billet were collected from”.

A6: Thanks for your suggestion. We have modified the sentence as follow: The planted-based meat samples, including beef jerky, vegetable sausage, beef grains, crab mayonnaise, and beef grains billet, were collected from the planted-based meat analogues company.

Q7. Line 92…. to collect the bacteria and fungi, respectively. In this sentence the term respectively indicates there is something missing. Please revise this sentence.

A7: Thanks for your correction. We have revised this sentence as follow: A 20 mL aliquot of the homogenized mixture was collected by centrifugation at 17,226 × g for 10 min to polish the separation of bacteria and fungi, respectively.

Q8. Line 92-93. bacterial genomic DNA was extracted using a bacterial genomic DNA extraction kit and fungal genomic DNA extraction, respectively. This statement is confusing, please revise.

A8: Thanks for your correction. We have modified this sentence as follow: bacterial and fungal genomic DNA was extracted using a bacterial genomic DNA ex-traction kit and fungal genomic DNA extraction, respectively.

Q9. Line 312 - 315. Redundancy of “The relative abundance of”. Please revise.

A9: Thanks for your suggestion. We have modified the sentence as follow: Figure 7. The relative abundance of aerobic, anaerobic, and facultative anaerobic bacteria. (A) The aerobic bacteria; (B) The anaerobic bacteria; (C) The facultative anaerobic bacteria;(D) The potential pathogens.

Q10. Please separate the conclusion part from the discussion, if possible, provide future directions along with it as well.

A10: Thanks for your suggestions. We have added the conclusion section as follow: In-depth analysis of the microbial composition of five types of plant-based meat analogues samples was performed using high-throughput sequencing technology. This analysis provides valuable information on the diversity and compositional changes of plant-based meat analogues. However, there are some areas for improvement in later study, such as more samples to ensure the accuracy of data, more kinds of plant-based meat analogues, the relationship between microbial composition and texture and pH value. Hence, further research directions include functional analysis of microbial communities associated with the raw materials and specific processing methods to prolong the product shelf-life, and reduced the proliferation of the pathogenic bacteria to ensure the food safety.